# Early Left Ventricular Diastolic Dysfunction in Females with Chronic Hyperprolactinemia: A Doppler Echocardiographic Study

**DOI:** 10.3390/jcm12041658

**Published:** 2023-02-19

**Authors:** Michele Arcopinto, Roberta D’Assante, Renata Simona Auriemma, Rosa Pirchio, Rosario Pivonello, Eduardo Bossone, Annamaria Colao, Antonio Cittadini

**Affiliations:** 1Department of Translational Medical Sciences, “Federico II” University of Naples, Via Sergio Pansini 5, 80131 Naples, Italy; 2Endocrinology Unit, Department of Clinical Medicine and Surgery, “Federico II” University of Naples, Via Sergio Pansini 5, 80131 Naples, Italy; 3Department of Public Health, “Federico II” University of Naples, Via Sergio Pansini 5, 80131 Naples, Italy

**Keywords:** hyperprolactinemia, prolactin, cardiac function, echocardiography

## Abstract

Despite the myocardial prolactin (PRL) binding activity and the known effect of enhancing contractility in the isolated rat heart, little information is available concerning the cardiovascular consequences of hyperprolactinemia in humans. To elucidate the effects of chronic hyperprolactinemia on cardiac structure and function, twenty-four patients with isolated PRL-secreting adenoma and twenty-four controls underwent a complete mono- and two-dimensional Doppler-echocardiography. Blood pressure and heart rate were similar in the two groups, and no significant differences were observed as to left ventricular (LV) geometry between patients and controls. Resting LV systolic function was normal in patients with hyperprolactinemia, as shown by similar values of fractional shortening and cardiac output. Conversely, hyperprolactinemic patients exhibited a slight impairment of LV diastolic filling, as demonstrated by the prolongation of the isovolumetric relaxation time and the increase of the atrial filling wave of mitral Doppler velocimetry (58 ± 13 vs. 47 ± 8 cm/s, *p* < 0.05) with a subgroup of females (16%) having a clear diastolic dysfunction, and a worse exercise capacity (6 min walking test 452 ± 70 vs. 524 ± 56; *p* < 0.05). In conclusion, hyperprolactinemia in humans may be associated with a slight impairment of diastolic function, with an overt diastolic dysfunction in a subgroup of females which correlated with poorer exercise performance, in the absence of significant abnormalities of LV structure and systolic function.

## 1. Introduction

Stimulation of lactation is classically described as the most relevant biological action of prolactin (PRL) [1]. However, the concept has recently emerged that PRL is a multifaceted hormone endowed with a broad variety of properties, encompassing growth stimulation, immunomodulation, neurotransmission, and regulation of water and electrolyte balance [1]. PRL may act as a hormone, by classical endocrine modality, as well as in a paracrine-autocrine fashion. It has been demonstrated that PRL has more actions than all pituitary hormones combined, including more than 300 separate functions already documented in vertebrates. Among these, PRL exerts a direct inotropic positive action on the mammalian myocardium, partly mediated via the local release of catecholamines and the cAMP pathway [2,3]. Furthermore, PRL increases the ornithine decarboxylase activity in mammalian hearts [4].

The demonstration of PRL binding sites by means of various techniques in both ventricular and atrial tissues provides a solid background to the hypothesis of a direct cardiovascular effect of PRL [5,6]. In myocardial infarction, circulating PRL levels are elevated and associated with an increased responsiveness of cardiovascular smooth muscle to norepinephrine and angiotensin and with a reduced threshold for electrical excitation [7,8]. Chronic hyperprolactinemia has been related to endothelial dysfunction and atherosclerosis [9] and its levels have been associated with a worse cardiovascular risk profile in post-menopausal women without prolactinoma [10]. More recently, PRL has been considered as a possible therapeutic target in peripartum cardiomyopathy, although with controversial benefits [11]. Despite evidence for a link between PRL and the heart, only a few systematic investigations have been so far performed in conditions of chronic PRL excess. This study was consequently undertaken to explore the impact of chronic PRL elevation on cardiac morphology and function in human subjects with PRL-secreting adenoma.

## 2. Materials and Methods

Twenty-four patients (five males and 19 females; mean age: 31 ± 7 yrs), with isolated PRL-secreting adenoma, were enrolled at Endocrinology Unit, Department of Clinical Medicine and Surgery, University of Naples “Federico II” from June 2021 to September 2022, after their informed consent was obtained. The diagnosis was based on the presence of plasma prolactin levels of at least 100 ng per ml and CT scan showing a pituitary adenoma. Eight out of 24 patients showed a macroadenoma. The average PRL concentration was calculated on the basis of 6-h time course with hourly sampling. Mean serum PRL concentration was 678 ± 996 ng/mL. Routine clinical and hormonal evaluations showed no evidence of any other pituitary hypersecretion and/or deficiency. None of the patients were treated with sexual steroids or bromocriptine before entering the study. Cardiac disease including systemic hypertension or coronary artery disease was excluded in all patients and all known cardiovascular risk factors were equally distributed in the two study groups.

Patients and control subjects underwent a standard 12-lead electrocardiogram and a complete mono- and two-dimensional Doppler-echocardiographic study. Complete M-mode and two-dimensional analyses were performed using an ultrasound mechanical system equipped with 3.5 mHz transducer (Apogee CX, Interspec, Ambler, CA, USA). M-mode and two-dimensional recordings were made with the subjects in a lateral recumbent position according to the standardizations of the American Society of Echocardiography [12]. All echocardiography studies were performed by the same investigator (A. C.), who was blinded to the subject group. Interventricular septal and left ventricular (LV) posterior wall thickness and LV end-diastolic and end-systolic cavity dimensions were measured by averaging the values of four consecutive cardiac cycles. As ejection phase indices, the percentage of LV fractional shortening was calculated as internal end-diastolic dimension minus internal end-systolic dimension divided by internal end-diastolic dimension multiplied by 100. LV ejection fraction was calculated according to the biplane method of disks (modified Simpson’s rule) as currently recommended [13]. LV mass was calculated using the following validated formula: Mass (grams) = 0.80 [1.04 × (LV internal end-diastolic dimension + ventricular septal thickness + posterior wall thickness) 3—(LV internal end-diastolic dimension) 3] + 0.6 [14]. The anteroposterior diameter of the left atrium was measured in the parasternal long-axis view perpendicular to the aortic root long axis and measured at the level of the aortic sinuses by using the leading-edge-to-leading-edge convention. Left atrium volume was measured in the four-chamber apical view, at end-systole, on the frame just prior to mitral valve opening, by tracing the LA inner border, excluding the area under the mitral valve annulus and the inlet of the pulmonary veins [13].

Doppler tracings were obtained during quiet respiration with the transducer positioned at the cardiac apical impulse and oriented to obtain an apical four-chamber view. The Doppler beam was aligned parallel to the presumed mitral inflow. The sample volume was placed at the inflow area of the left ventricle just below the mitral valve annulus, and its position was optimized to obtain maximal diastolic flow velocities and a better-defined Doppler waveform. The time interval from aortic valve closure to the onset of mitral early diastolic flow or isovolumic relaxation time (IRT) was determined as index of LV filling by simultaneous recording of the aortic and mitral flow by continuous-wave Doppler. Tissue Doppler was used to determine e′ at lateral and septal basal segments of LV. Left ventricular diastolic function was evaluated as recently recommended [13]. In our patients with a normal LV ejection fraction, diastolic function was also assessed considering the following additional parameters and cut-offs: annular E’ velocity (septal e’ < 7 cm/s, lateral e’ < 10 cm/s), average E/e’ ratio > 14, LA volume index > 34 mL/m^2^, and peak TR velocity > 2.8 m/s. Diastolic function was established as follows: normal diastolic function with no or one abnormal parameters; undetermined diastolic function with two abnormal parameters; diastolic dysfunction with three or four parameters. A six-minute walking distance (6-MWD) test was obtained for all participants.

Serum PRL levels were assessed by RIA using commercial kits (Radim, Pomezia, Italy). The intra- and inter-assay coefficients of variation for PRL were 5% and 7%, respectively. The normal range was below 20 ng/mL. Serum estradiol levels were assessed by RIA using commercial kits (Cis, Gif-Sur-Yvette Cedex, France). The intra- and inter-assay coefficients of variation for estrogen were 3.5% and 4.8%, respectively. In females, the normal range was 25–100 pg/mL during the follicular phase; in males, the normal range was 15–60 pg/mL. Serum progesterone levels were assessed by RIA using commercial kits (Cis, Gif-Sur-Yvette Cedex, France). The intra- and inter-assay coefficients of variation for progesterone were 7% and 9%, respectively. In females, the normal range was 0.2–1.0 ng/mL during the follicular phase; in males, the normal range was 0.2–0.8 ng/mL. Serum FSH levels were assessed by IRMA using commercial kits (Cis, Gif-Sur-Yvette Cedex, France). The intra- and inter-assay coefficients of variation for FSH were 4.7% and 7.4%, respectively. The normal range was 3–11 U/mL in females during the follicular phase and 1.5–7 U/mL in males. Serum LH levels were assessed by IRMA using commercial kits (Cis, Gif-Sur-Yvette Cedex, France). The intra- and inter-assay coefficients of variation for LH were 1.8% and 4.8%, respectively. The normal range was 0.5–12 U/mL in females during the follicular phase and 1.1–11.7 U/mL in males. Serum androstenedione levels were assessed by RIA using commercial kits (Radim, Pomezia, Italy). The intra- and inter-assay coefficients of variation for androstenedione were 5% and 8%, respectively. The normal range was 1.2–2.5 ng/mL.

The study protocol was approved by the Ethics Committee of the Federico II University of Naples (Prot. *n*. 245/21).

All values are presented as mean ± SD. Statistical analysis was performed by Statview statistical software. After testing for normal distribution, comparison between the two study groups was performed with the unpaired two-tailed Student’s t-test with Bonferroni correction. Corresponding non-parametric tests were used as appropriate. Linear regression analysis was used to determine whether correlation existed between variables. A *p* value below 0.05 was considered significant.

## 3. Results

The clinical and hormonal data of the 24 patients and the 24 sex- and age-matched control subjects are shown in Table 1.

In addition to the increased serum levels of PRL, all patients showed a reduction of gonadotropins levels and female patients displayed a significant reduction of circulating estradiol levels (Table 2).

Electrocardiograms were normal in all but one female patient, who showed mild repolarization abnormalities. No significant differences between patients and controls were observed concerning blood pressure, heart rate, and LV morphology, as summarized in Table 3.

In particular, LV mass (g) was 163 ± 52 vs. 169 ± 39 (*p* = NS); the end-diastolic dimension (mm) was 48 ± 3 vs. 48 ± 2 (*p* = NS); the end-systolic dimension (mm) was 29 ± 3 vs. 29 ± 3 (*p* = NS); there was a slight, but not significant, increase of left atrial anteroposterior diameter and indexes volume in the PRL group. LV systolic function was not altered in chronic PRL excess, as shown by similar values of ejection phase indexes between the two study groups (LV ejection fraction (%): 61 ± 3 vs. 62 ± 2, *p* = NS; FS: 39 ± 5 vs. 39 ± 5, *p* = NS; CO (L/min): 5.5 ± 10 vs. 5.3 ± 1, *p* = NS). On the contrary, the two groups displayed a significant difference in the LV diastolic filling pattern with mitral Doppler velocimetry in PRL groups showing an increased A wave (cm/sec) (58.52 ± 13.19 vs. 47.73 ± 7.90, *p* < 0.05), which represents the atrial contribution to LV filling. Consequently, the early-to-late filling velocity (E/A) ratio was lower in the PRL group (1.44 ± 0.46 vs. 1.66 ± 0.25, *p* < 0.05) suggesting an early impairment of LV relaxation pattern (Table 3). Such finding was further supported by a 19% prolongation of the isovolumetric relaxation time. However, only three subjects (12.5%) in the PRL group (all females) displayed a defined diastolic dysfunction according to current definition (16% of females with PRL) while seven subjects displayed an undetermined diastolic function (29%, six females, one male). In the control group, no subjects showed diastolic dysfunction expect one with undetermined function. No correlation was found between mitral E/A ratio, E/E’ ratio, diastolic function status and any of the hormonal measurements. Hormonal levels in both sexes are shown in Table 3. The 6-MWD test was slightly lower in the PRL group compared to controls (515 ± 60 m vs. 610 ± 54 m, *p* < 0.05). A more pronounced reduction of 6-MWD was observed in PRL individuals with diastolic dysfunction compared with those showing normal or undetermined function (452 ± 70 vs. 524 ± 56, *p* < 0.05), which was about 26% lower than mean 6-MWD in controls.

## 4. Discussion

The current study demonstrates that chronic hyperprolactinemia in humans does not cause any significant change in LV morphology and systolic function evaluated with LV fractional shortening, LV ejection fraction or cardiac output. A mild, preclinical impairment of LV systolic function has been recently described only with a Tissue Doppler approach [15]. However, LV diastolic filling and exercise capacity are significantly impaired in PRL subjects, with a small proportion of females with PRL showing overt diastolic dysfunction. Although the main target organ of PRL is the breast, specific lactogenic receptors have been identified in other tissues, such as liver, kidney, adrenals, ovaries, testis, prostate, etc. [5]. The receptor density is variable as a function of the estrous cycle, pregnancy, and lactation [1]. Ventricular and atrial tissues have also been shown to contain PRL binding sites, although their exact role is unclear. Ouhtit et al. have demonstrated the expression of a long form of PRL receptor (PRLR) in the rat heart [6]. In addition, Freemark et al. documented the appearance of PRLR in cardiac myocytes of the human foetus [16]. Early animal studies have consistently reported that PRL has a positive inotropic action on isolated rat and rabbit heart preparations, although the mechanism(s) of action have not been elucidated yet [2,3]. It has been speculated that this inotropic effect may be mediated by endogenous catecholamine release since it was significantly blunted by indomethacin and propanolol [2]. There were no differences concerning LV mass between hyperprolactinemic patients and controls, which allows us to exclude a trophic role of PRL on the human myocardium. Furthermore, PRL-mediated positive inotropy was not demonstrable by echocardiographic-derived systolic phase indexes, such as fractional shortening, ejection fraction, or aortic flow velocity. We do not know the potential time of exposure to hyperprolactinemia in the study group. Thus, one potential explanation for the absence of differences between groups could well be a limited time of exposure. Should the patients with hyperprolactinemia have had a short clinical course from the onset of the symptoms (presumably the time when the levels of prolactin became significantly high) to the time of the study measurements, we would not really expect an important change in cardiac structure and function. On the other hand, diastolic filling was impaired in hyperprolactinemia, as shown by the prolongation of the isovolumetric relaxation time, a decrease of atrial peak flow velocity, and a slight increase of left atrial dimension, all pointing to a delayed relaxation pattern [17]. Such alterations were even more pronounced in a small proportion of PRL subjects, showing more than two out of four ultrasound characteristics of diastolic dysfunction (12.5%) with only about 42% showing none of them, compared with 95.8% in the control group. Such subjects represent the ones with the worse performance at 6-MWD test; the relation between diastolic dysfunction and poorer exercise performance is well established and may also be at play in this clinical condition [18]. Impaired diastolic filling usually precedes systolic dysfunction and is the first sign of cardiac involvement in hypertension, myocardial ischemia, and other cardiovascular diseases [17]. Diastolic relaxation is more susceptible than systolic contraction to an imbalance between energy supply and utilization. Myocardial relaxation requires Ca++ sequestration by the sarcoplasmic reticulum, which is an active ATP-dependent process (14). Small decreases in ATP availabilities can impair relaxation by the weakened allosteric effect of ATP to inhibit the actin-myosin interaction. However, only major decreases in ATP levels will lead to contractile abnormalities (14).

The clinical nature of the current study does not allow definite conclusions to be drawn as to the exact mechanism(s) underlying the impaired diastolic filling in hyperprolactinemia. It is possible that PRL excess per se impairs relaxation by interfering with the rates of Ca++ sequestration by the sarcoplasmic reticulum, possibly reducing ATP disposal. Alternatively, indirect mechanisms might be at play, such as the hypoestrogenism induced by hyperprolactinemia. It is well accepted that estrogens target the cardiovascular system also independently of the known effects on lipid metabolism and have significant impacts on diastolic function. Long-term estrogen deprivation alters LV filling and function and decreases cardiac myosin ATPase activity in adult rats [19]. Conversely, estrogen replacement therapy improves some parameters of diastolic function in postmenopausal women, particularly the mitral E/A ratio [20]. It is possible that the relaxation abnormalities documented in our patients may depend upon the hypoestrogenic state induced by chronic PRL excess, with the attendant reduction of myosin ATPase activity and the impairment of Ca++ reuptake by the sarcoplasmic reticulum. This speculation is supported by the lack of diastolic abnormalities in the five hyperprolactinemic males included in the study. Supportive evidence comes from a recent work by Michalson et al. which demonstrated an enhanced LV diastolic function and altered myocardial gene expression towards decreased extracellular matrix deposition and calcium homeostasis in an animal model of ovariectomy with subsequent estradiol supplementation, suggesting that estradiol directly or indirectly modulates the myocardial transcriptome to preserve cardiovascular function [21].

## 5. Conclusions

The clinical relevance of an impaired diastolic function in hyperprolactinemic humans is unclear. For example, it is of interest to see whether more overt signs of cardiac involvement appear later during the disease course. Further longitudinal evaluation is needed to clarify this issue and to assess the efficacy of therapeutic interventions on the reversal of cardiac abnormalities in hyperprolactinemia.

## Figures and Tables

**Table 1 jcm-12-01658-t001:** Clinical and hormonal characteristics of the study population.

	Hyperprolactinemia *n* = 24	Control *n* = 24
Sex (M/F)	5/19	5/19
Age (yrs)	31 ± 7	30 ± 3
Height (cm)	162 ± 5	164 ± 10
Weight (kg)	72 ± 14	71 ± 13
Body Surface Area (m^2^)	1.77 ± 0.16	1.75 ± 0.19
Heart Rate (bpm)	72 ± 11	71 ± 9
Mean Blood Pressure (mmHg)	88 ± 8	85 ± 8
PRL (ng/mL)	678 ± 996 *	15.6 ± 3.2
Estradiol (pg/mL)	40 ± 21 *	81 ± 49
FSH (U/mL)	3.4 ± 2.4 *	9.9 ± 3.2
LH (U/mL)	2.8 ± 3 *	9.6 ± 2.2
Androstenedione (ng/mL)	2.9 ± 0.8	2.3 ± 1
Progesterone (ng/mL)	1.3 ± 0.5	2.0 ± 0.6

PRL = prolactin; FSH= Follicle-Stimulating Hormone; LH = Luteinizing hormone. * = *p* < 0.05 vs. controls.

**Table 2 jcm-12-01658-t002:** Hormonal characteristics of males and female hyperprolactinemic patients.

	Females *n* = 19	Males *n* = 5
PRL (ng/mL)	607 ± 889	580 ± 302
Estradiol (pg/mL)	39 ± 20 *	20 ± 2
FSH (U/mL)	3.3 ± 2.2	4.5 ± 3.5
LH (U/mL)	2.9 ± 3	2.3 ± 1.4
Androstenedione (ng/mL)	2.9 ± 0.8	3.1 ± 0.3
Progesterone (ng/mL)	1.2 ± 0.5	0.5 ± 0.3

PRL = prolactin; FSH= Follicle-Stimulating Hormone; LH = Luteinizing hormone. * = *p* < 0.05 vs. controls.

**Table 3 jcm-12-01658-t003:** Echocardiographic parameters and physical performance in controls and hyperprolactinemic subjects.

	Hyperprolactinemia	Control
	*n* = 24	*n* = 24
Aortic root (mm)	30 ± 3	28 ± 3
Left atrium volume index (mL/m^2^)	27 ± 7 *	22 ± 6
IS diastole (mm)	9 ± 2	9 ± 1
PW diastole (mm)	8 ± 7	8 ± 1
LV-EDD (mm)	48 ± 3	48 ± 3
LV-ESD (mm)	29 ± 3	29 ± 3
LV Mass (g)	163 ± 52	169 ± 40
Fractional shortening (%)	39 ± 5	39 ± 5
LV ejection fraction (%)	61 ± 3	60 ± 2
Cardiac output (l/min)	5.3 ± 1	5.5 ± 1
TAPSE (mm)	22 ± 4 *	25 ± 3.6
Peak TR velocity (m/s)	2.5 ± 0.6 *	2.1 ± 0.4
IRT (ms)	93 ± 15	75 ± 9
Mitral DT (ms)	147 ± 28	142 ± 27
Mitral E/A ratio	1.44 ± 0.46 *	1.66 ± 0.25
Mitral E/E’ ratio	7.8 ± 1.9 *	6.4 ± 2.0
LV diastolic function:		
normal diastolic function (*n*, %)	14 (58.4) *	23 (95.8)
undetermined function (*n*, %)	7 (29.1) *	1 (4.2)
diastolic dysfunction (*n*, %)	3 (12.5) *	0 (0)
6 MWD (m)	515 ± 60 *	610 ± 54

IS = interventricular septum; PW = posterior wall; LV = left ventricular; EDD = end-diastolic dimension; ESD = end-systolic dimension; TAPSE = Tricuspid annular plane systolic excursion; TR = Tricuspid regurgitation; IRT = isovolumic relaxation time; DT = deceleration time; E = mitral early peak flow velocity; A = mitral late peak flow velocity; E’: mitral annular early diastolic velocity; 6 MWD: 6 min walking distance; * = *p* < 0.05 vs. controls.

## Data Availability

The data presented in this study are available on request from the corresponding author. The data are not publicly available due to privacy restrictions.

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
