# Peer review of "Early Left Ventricular Diastolic Dysfunction in Females with Chronic Hyperprolactinemia: A Doppler Echocardiographic Study"

_jcm, 2023, doi:10.3390/jcm12041658_

Round 1
Reviewer 1 Report
Early Left Ventricular Diastolic Dysfunction in Females with Chronic Hyperprolactinemia: a Doppler Echocardiographic Study
Manuscript entitled “Early Left Ventricular Diastolic Dysfunction in Females with Chronic Hyperprolactinemia: a Doppler Echocardiographic Study” by Michele et al., is a good study. Here, the authors showed the effects of chronic hyperprolactinemia on cardiac structure and function in patients with isolated PRL-secreting adenoma using two-dimensional Doppler-echocardiography. The author demonstrated that the hyperprolactinemia in humans maybe associated with slight impairment of diastolic function, with an overt diastolic dysfunction in a subgroup of females which correlated with poorer exercise performance, in the absence of significant abnormalities of LV structure and systolic function.
Overall, the information presented in this manuscript a useful information and I approve its publication after some minor updates.
Minor comments: I suggest that these comments to be updated before publication.
§ --- Introduction part should be improved by including more information about Chronic Hyperprolactinemia.
§ --- In Methods section – Information about the hospital, ethical approval and year of sample collection is not mentioned.
§ --- Conclusion part should be improved.
Author Response
Manuscript entitled “Early Left Ventricular Diastolic Dysfunction in Females with Chronic Hyperprolactinemia: a Doppler Echocardiographic Study” by Michele et al., is a good study. Here, the authors showed the effects of chronic hyperprolactinemia on cardiac structure and function in patients with isolated PRL-secreting adenoma using two-dimensional Doppler-echocardiography. The author demonstrated that the hyperprolactinemia in humans maybe associated with slight impairment of diastolic function, with an overt diastolic dysfunction in a subgroup of females which correlated with poorer exercise performance, in the absence of significant abnormalities of LV structure and systolic function.
Overall, the information presented in this manuscript a useful information and I approve its publication after some minor updates.
Minor comments: I suggest that these comments to be updated before publication.
- Introduction part should be improved by including more information about Chronic Hyperprolactinemia.
- We accept the reviewer's comment and have improved the Introduction section on chronic hyperprolactinemia, particularly in relation to the etiology and clinical presentation. Please see page 2, lines 51-59.
- In Methods section – Information about the hospital, ethical approval and year of sample collection is not mentioned.
- We thank the reviewer for the comment and as suggested, we added these information in Materials and Methods section (page 2, lines 70-71, page 3 lines 150-151).
- Conclusion part should be improved.
- We accept the reviewer's comment and have improved the conclusion section, particularly on the possible clinical implications. Please see page 7, lines 290-292.
Reviewer 2 Report
The manuscript entitled "Early Left Ventricular Diastolic Dysfunction in Females with Chronic Hyperprolactinemia: a Doppler Echocardiographic Study" explores the effect of increased prolactin levels on cardiac structure and function. The authors need to address the following queries
1. The authors need to explain the greater standard deviation of Prolactin levels compared to mean values for the study group
2. Under section 3 results, Table 1 the M/F for Hyperprolactinemia group is mentioned as 19/5, while it should have been 5/19
3. Under section 2 Materials and Methods "Left ventricular diastolic function was evaluated as recently recommended (ref)" reference has not been cited
4. Under results section 3, in most of the instances mean and SD deviation have been expressed as mean+SD
5. Assessment of Left ventricular diastolic dysfunction can be elaborated under the Materials and Methods section.
6. Under section 2 Materials and Methods, "The normal range was below 20 g/L", verify the unit of Prolactin levels
7. Footnotes for tables (table 2 and table 3) can be corrected
8. Under the Discussion section "It is possible that the relaxation abnormalities documented in our patients may depend upon the hypoestrogenic state induced by chronic PRL excess, with the attendant reduction of myosin ATPase activity and the impairment of Ca++ reuptake by the sarcoplasmic reticulum". However, the results demonstrate that although the estrogen level in the study group is lower than the control group, it is still within the normal range. Justify
9. Rectify the typographical errors
Author Response
The manuscript entitled "Early Left Ventricular Diastolic Dysfunction in Females with Chronic Hyperprolactinemia: a Doppler Echocardiographic Study" explores the effect of increased prolactin levels on cardiac structure and function. The authors need to address the following queries
- The authors need to explain the greater standard deviation of Prolactin levels compared to mean values for the study group
R: The high standard deviation of prolactin levels in the study group is attributable to the high variability of PRL levels at the time of diagnosis. The finding was also noticed in previous studies or case series because PRL levels are strictly related to the pituitary adenoma size (Bianca M Leca , Maria Mytilinaiou , Marina Tsoli , Andreea Epure , Simon J B Aylwin , Gregory Kaltsas , Harpal S Randeva , Georgios K Dimitriadis. Identification of an optimal prolactin threshold to determine prolactinoma size using receiver operating characteristic analysis. Sci Rep. 2021 May 7;11(1):9801.doi: 10.1038/s41598-021-89256-7). No restrictive criteria regarding the adenoma dimensions were used in this study (inclusion criteria: diagnosis of prolactinoma), thus the PRL levels in affected subjects was expected to be extremely heterogeneous.
- Under section 3 results, Table 1 the M/F for Hyperprolactinemia group is mentioned as 19/5, while it should have been 5/19
R: We apologize for the mistake. We changed the data accordingly.
- Under section 2 Materials and Methods "Left ventricular diastolic function was evaluated as recently recommended (ref)" reference has not been cited
R: We apologize for the mistake. The missing reference has been added. Please see page 3, line 122.
- Under results section 3, in most of the instances mean and SD deviation have been expressed as mean+SD
R: We apologize for the mistake that has been carefully checked and corrected throughout the text
- Assessment of Left ventricular diastolic dysfunction can be elaborated under the Materials and Methods section.
R: The assessment of left ventricular diastolic dysfunction is actually elaborated under the materials and methods section.
- Under section 2 Materials and Methods, "The normal range was below 20 g/L", verify the unit of Prolactin levels
R: We apologize for the mistake. Unit of prolactin levels have been corrected with the exact one: ng/mL.
- Footnotes for tables (table 2 and table 3) can be corrected
R: We thank the reviewer for his comment which allowed us to implement in the correct way the footnotes. We added more details regarding the variables cited in table 2 and 3.
- Under the Discussion section "It is possible that the relaxation abnormalities documented in our patients may depend upon the hypoestrogenic state induced by chronic PRL excess, with the attendant reduction of myosin ATPase activity and the impairment of Ca++ reuptake by the sarcoplasmic reticulum". However, the results demonstrate that although the estrogen level in the study group is lower than the control group, it is still within the normal range. Justify
R: The point raised by the Reviewer is correct. We described the relation between this relative hypoestrogenic state and mild diastolic dysfunction as “possible”, as we do not have definitive data about it. However, there are solid evidence about this relation in the animal model. We updated the text acknowledging the Reviewer comment. Please see page 7 Line 270
- Rectify the typographical errors
R: We apologize for typo errors and we did our best to correct them.
Reviewer 3 Report
Relevant topic to investigate, but the sample size is too small, and the findings are also not strong enough. Does this study have IRB / Ethical approval, if yes then the author must add those in the methods section.
Author Response
REVIEWER 3
Relevant topic to investigate, but the sample size is too small, and the findings are also not strong enough. Does this study have IRB / Ethical approval, if yes then the author must add those in the methods section.
R: We understand the Reviewer comment; a larger sample of patients would certainly have given more strength to the study to better characterize the mild abnormalities we found. We updated the text acknowledging the Reviewer comment in the conclusion section (page 7, lines 290-294). Regarding the Ethical approval, we added this information in Materials and Methods section (page 3, lines 150-151).
Round 2
Reviewer 2 Report
The authors have addressed all the comments
Reviewer 3 Report
I think the author have significantly improved the issues